# Prospective Analysis of B Lymphocyte Subtypes, before and after Initiation of Dialysis, in Patients with End-Stage Renal Disease

**DOI:** 10.3390/life13040860

**Published:** 2023-03-23

**Authors:** Dimitra-Vasilia Daikidou, Georgios Lioulios, Erasmia Sampani, Aliki Xochelli, Vasiliki Nikolaidou, Eleni Moysidou, Michalis Christodoulou, Artemis Iosifidou, Myrto Iosifidou, Dimitria Ioanna Briza, Aikaterini Papagianni, Asimina Fylaktou, Maria Stangou

**Affiliations:** 1School of Medicine, Aristotle University of Thessaloniki, 45636 Thesaloniki, Greece; 2Department of Nephrology, Hippokration Hospital, 54642 Thessaloniki, Greece; 3Department of Immunology, National Histocompatibility Center, Hippokration General Hospital, 54642 Thessaloniki, Greece; 4School of Informatics, Aristotle University of Thessaloniki, 45636 Thesaloniki, Greece

**Keywords:** B lymphocytes, end-stage renal disease, hemodialysis, peritoneal dialysis, apoptosis, innate B1 cells, conventional B2 cells

## Abstract

End-stage renal disease (ESRD) is followed by alterations in adaptive immunity. The aim of this study was to evaluate B lymphocyte subtypes in ESRD patients before and after hemodialysis (HD) or continuous ambulatory peritoneal dialysis (CAPD). Patients and Methods. CD5, CD27, BAFF, IgM and annexin were evaluated by flow cytometry on CD19+ cells in ESRD patients (n = 40), at time of initiating HD or CAPD (T0) and 6 months later (T6). Results. A significant reduction in ESRD-T0 compared to controls was noticed for CD19+, 70.8 (46.5) vs. 171 (249), *p* < 0.0001, CD19+CD5−, 68.6 (43) vs. 168.9 (106), *p* < 0.0001, CD19+CD27−, 31.2 (22.1) vs. 59.7 (88.4), *p* < 0.0001, CD19+CD27+, 42.1 (63.6) vs. 84.3 (78.1), *p* = 0.002, CD19+BAFF+, 59.7 (37.8) vs. 127.9 (123.7), *p* < 0.0001 and CD19+IgM+ cells, 48.9 (42.8) vs. 112.5 (81.7) (K/μL), *p* < 0.0001. The ratio of early/late apoptotic B lymphocytes was reduced (16.8 (10.9) vs. 110 (25.4), *p* = 0.03). CD19+CD5+ cells were the only cell type with an increased proportion in ESRD-T0 patients (2.7 (3.7) vs. 0.6 (1.1), *p* < 0.0001). After 6 months on CAPD or HD, CD19+CD27−(%) and early apoptotic lymphocytes were reduced further. The HD patients also showed a significant increase in late apoptotic lymphocytes, from 1.2 (5.7) to 4.2 (7.2) K/mL, *p* = 0.02. Conclusions. B cells and most of their subtypes were significantly reduced in ESRD-T0 patients compared to controls, the only exception being CD19+CD5+ cells. Apoptotic changes were prominent in ESRD-T0 patients and were exacerbated by HD.

## 1. Introduction

During the gradual deterioration of the kidney function towards End-Stage Renal Disease (ESRD), several clinical complications arise, such as malignancies, increased incidence of cardiovascular diseases, atheromatosis, susceptibility to infections, reduced response to vaccination, etc. [1,2,3,4]. Most of these situations are the direct effect of kidney dysfunction, which seems to act as a sustained chronic inflammatory disease, yet, at the same time is characterized by unique features, such as uremia, uremic toxic as well as O_2_ free radicals and advanced glycation end products accumulation [1,3,5,6,7]. 

ESRD has a specific and detrimental effect on immune integrity, affecting both innate and adaptive immunity. Innate immunity, the first line of defense, depends mainly on cell function and also on the recognition of pathogen- and damage-associated molecular patterns (PAMPs, and DAMPs) [1,8,9]. Adaptive immunity is directed by T and B lymphocytes, develops following the recognition of a specific antigen and aims to detect, defend and furthermore develop memory adaptive mechanisms against any pathogen.

Although ESRD affects the number and function of neutrophils, macrophages and surface antigens, its main detrimental effects are connected to the adaptive immune compartment [10]. ESRD has been proved to cause particular and unique effects on lymphocyte phenotype and function, characterized as an ESRD-lymphocyte pattern. In our previous studies, we extensively described alterations of T lymphocytes and, correspondingly, studied the effect of different dialysis methods on the T cell phenotype [11,12,13]. Our previous findings were indicative of a divergent effect of two dialysis methods, i.e., Hemodialysis (HD) and Continuous Ambulatory Peritoneal Dialysis (CAPD), on the surface receptors of T lymphocytes, favoring CAPD, as it appeared to be associated with less senescent T cell alterations, better approaching the normal phenotype. We also analyzed the possible re-instatement after a successful renal transplantation and proved that some of T cell markers were restored, while others remained unaltered after kidney transplantation [14,15].

In this prospective study, we assessed the ESRD-associated phenotypic changes in the B lymphocyte compartment, as they presented initially at a pre-dialysis stage and progressed after the patient was started on a dialysis method. Therefore, we analyzed B lymphocytes and their subpopulations in a cohort of ESRD patients at two time points: the day they started on a dialysis method and 6 months later, while being on either HD or CAPD, under stable conditions. The first measurement was representative of the effect of ESRD at the pre-dialysis stage on the B lymphocyte compartment, while the second assessment, following a 6 month period on a dialysis method, displayed the further effect of dialysis on B cells. Furthermore, we compared the effects of HD and CAPD on B lymphocyte phenotypic alterations.

## 2. Materials and Methods

### 2.1. Patients

In the present study, we included patients with ESRD at the pre-dialysis stage and prospectively followed them until 6 months after commencing on a dialysis method, either HD or CAPD.

All patients were European Caucasians, aged 18–75 years, and they all had been regularly followed in the “Chronic Kidney Disease” outpatients clinic for at least 12 months prior to enrollment. During this period, they received adequate treatment of co-morbid conditions and complications due to kidney failure, such as hypertension, anemia, secondary hyperparathyroidism. The deterioration of the renal function was gradual, with no evidence of acute on chronic kidney disease.

Exclusion criteria were active systemic disease, malignancy or hematological disorder, recent, during the last 6 months, bacterial or viral infection, treatment with corticosteroids or immunosuppressives, vaccination for COVID-19, influenza virus, herpes zoster or pneumococcus. Patients who had received monoclonal antibodies against B lymphocytes—Rituximab or Belimumab—were also excluded, regardless of the time they had received this treatment.

Healthy volunteers, matched for age, sex and race, served as controls.

Forty-five ERSD patients, who fulfilled the above criteria, were initially included. The study protocol was approved by the Ethics Committee of Hippokration Hospital of Thessaloniki and followed the general principles of the Declaration of Helsinki (2008 amendment). All subjects provided informed consent prior to study enrollment.

### 2.2. Schedule of the Study

The study was conducted in two stages.

During the first stage, the B cell phenotype was assessed in order to compare the B cell immunity between ESRD patients, before starting on dialysis, and healthy controls (HC). The time of enrollment was defined as T0 and was the day that patients with ESRD started on either HD or CAPD. B cell immunity was estimated based on the phenotypic alterations of B lymphocytes, namely, the expression of receptors, characteristic for specific B cell subpopulations. At the same time point, T0, demographic, anthropometric data and clinical characteristics were recorded for both patients and controls. The patients’ laboratory assessment included the estimation of their hematological, biochemical and immunological profile, comprising indices of chronic inflammation, C-reactive protein (CRP) and serum albumin levels, secondary hyperparathyroidism, serum Calcium (Ca), serum Phosphate (P), intact parathyroid hormone (PTH) and serum immunoglobulin levels (IgA, IgG and IgM).

During the second phase of the study, a prospective follow-up of the patients’ immune profile was conducted, after the patients started on either HD or CAPD. All patients were eligible for both treatment options, and the decision was made upon their preference and for socioeconomic reasons.

The patients were prospectively followed for 6 months. During this period, they were monitored for stability in terms of cardiovascular complications, treatment of anemia and secondary hyperparathyroidism. Sessions for the HD patients were performed three times per week, in alternative days, with at least 4 h per session and using biocompatible membranes. The patients on CAPD did not have to present surgical complications due to catheter insertion, peritonitis episodes or catheter infections during the follow-up period and also had to be adequately treated for chronic kidney disease (CKD) complications.

Six months after commencing on dialysis, at time point T6, the patients were reevaluated in terms of the B lymphocyte phenotypic alterations happening while they had been either on HD or on CAPD.

The first blood samples, at T0, were collected either just before the first HD session or just before starting regularly on the peritoneal exchanges for the HD and CAPD patients, respectively. The follow-up samples, at T6, were collected at the beginning of a mid-week dialysis session for the HD patients. The patients are defined as ESRD-T0 and ESRD-T6 at the beginning of the study and the end of the follow-up, respectively.

Patients were withdrawn from the study if during the 6 months, they were lost of follow-up, had a systemic bacterial infection, CMV or corona virus infection, underwent any vaccination, were diagnosed with malignancy, needed to transfer to an alternative dialysis method, could not achieve dialysis adequacy and finally if they were uncompliant. The adequacy of dialysis was assessed at T6, by KT/V calculation. HD patients with KT/V ≤ 1.2 per session or CAPD patients with KT/V ≤ 1.7 per week were withdrawn from the study, as well as patients who had complications linked to the arterial–venous fistula (AVF) or CAPD catheter.

### 2.3. B Cell Immunity and Subpopulations

The number of peripheral B lymphocytes was calculated based on the presence of CD19+ receptors, CD5, BAFF, IgM and CD27 and included innate B1 (CD19+CD5+), conventional or B2 (CD19+CD5−), naïve (CD19+CD27−), proliferative (CD19+BAFF+), non-switched (CD19+IgM+) and memory (CD19+CD27+) cells.

The presence of annexin on T lymphocytes and CD19+ lymphocytes was used to define apoptotic lymphocytes and apoptotic B cells. We further distinguished apoptotic cells as early and late apoptotic, based on the density of annexin expression.

### 2.4. Laboratory Measurements

#### Flow Cytometry

Whole blood was drawn from healthy controls at T0 and from ESRD patients at T0 and T6. Blood samples were collected in EDTA tubes and processed for lymphocyte count and evaluation of B lymphocytes and subpopulations after no more than 3 h from blood collection. The total number of white blood cells (CD45+ cells) and subtypes, including T lymphocytes (CD3+ cells), was determined with a cell counter (Navios Flow Cytometer, Beckman Coulter). The antibody used for the detection and calculation of B lymphocytes was CD19+ PC5 (Beckman Coulter) and those for the measurements of subpopulations were CD5 PE (Clone L17F12, EXBIO), CD27 APC (Clone LT27 EXBIO), BAFF PC7 (CD268, Biolegent), IgM FITC (RabbitF(ab)’2, Dako), ApoFlowEx Annexin V FITC (EXBIO); Propidium Iodine (PI, EXBIO) was also used. Based on the presence of the above surface receptors, the B cell subtypes defined were innate B1 cells (CD19+CD5+), conventional B2 cells (CD19+CD5−), naïve B cells (CD19+CD27−), non-switched B cells (CD19+IgM+), memory cells (CD19+CD27+); proliferative cells (CD19+BAFF) and apoptotic cells were evaluated. The gating strategy and the isolation of singlets are described in Appendix A.

The percentage of CD19+ cells in lymphocytes was estimated, while the percentages of CD19+CD5+, CD19+27−, CD19+IgM, CD19+CD27+, CD19+BAFF+ were evaluated in CD19+ cells. The percentage of both apoptotic lymphocytes and apoptotic B cells were estimated.

### 2.5. Statistics

Statistical analysis was performed with the Statistical Package for Social Sciences (SPSS Inc., Chicago, IL, USA) version 25.0 for windows. The tests Kolmogorov–Smirnov and/or Shapiro–Wilk were used to assess the distribution of continuous variables. The data of normally distributed variables were expressed as Mean ± Standard Deviation, and the data of non-parametric variables as Medians and Interquartile Range (IQR). Student’s *t* test for non-paired samples and Mann–Whitney U test were performed to compare differences between groups for normally distributed and non-parametric variables, respectively. Similarly, Student’s *t* test for paired samples and Wilcoxon test were performed to assess differences in data before and after the treatments. Pearson’s and Spearman’s coefficients were used for the correlation between parametric and non-parametric variables respectively. Values of *p* < 0.05 (two-tailed) were considered statistically significant for all comparisons.

## 3. Results

Forty-five patients were initially evaluated, provided their consented and were included in the study. Five of them were withdrawn in the first month, one because of recurrent infections, one because of death due to cardiac arrest, and three because they moved home and were lost to follow-up.

The primary diseases in the cohort of 40 patients who remained in the study were diabetic nephropathy, diagnosed in 12 patients (30%), hypertension, in 11 (27.5%), primary glomerulonephritis, in 8 (20%), polycystic kidney disease, in 4 (10%) and other diseases in 5 (12.5%). According to the protocol, no patient had received steroids or immunosuppressive treatment or had had a systemic disease or infection during the last 6 months.

### 3.1. Differences between Patients and Controls

Table 1 and Table 2 summarize the hematological differences, including lymphocyte count, biochemical differences and differences in chronic inflammatory markers and immunoglobulin profiles, between patients and controls.

Our ESRD-T0 patients had total white cell counts with similar to those of the healthy controls, but there was a significant disturbance in the balance between neutrophils and lymphocytes, with a clear shift towards neutrophils and a reduction in lymphocytes, resulting in a significant increase of the Neutrophil/Lymphocyte Ratio (NLR) in ESRD-T0 patients. No significant differences were evident regarding the rest of myeloid cells and also the levels of immunoglobulins.

#### Differences in B Lymphocytes and Their Subpopulations between ESRD-T0 Patients and Healthy Controls

The different expression of B cell phenotypes is presented at Table 3 and depicted in Figure 1. There was an extremely significant reduction of the B lymphocyte compartment and its subpopulations in the ESRD-T0 patients compared to the controls. Most interestingly, the only B cell type which was increased in the ESRD-T0 patients was that of innate B1 cells, with a proportion rising to almost four times, compared to the controls. Although the proportions of the rest of the B cell subpopulations studied were similar in ESRD-T0 patients and controls, the absolute numbers of all subtypes, including naïve, memory, non-switched and CD19+BAFF+ cells, were significantly reduced in the ESRD-T0 patients compared to the controls.

### 3.2. Apoptotic Indices in Total Lymphocytes and B Lymphocytes

The apoptotic phenomenon in CD3+ lymphocytes and B lymphocytes showed substantial differences between ESRD-T0 patients and controls. The percentage of early apoptotic lymphocytes was similar; however, the number of the late apoptotic lymphocytes was significantly increased in ESRD-T0 patients compared to healthy controls. (Table 4, Figure 2). The ratio of early/late apoptotic cells was significantly reduced in ESRD-T0 patients, demonstrating a shift to late apoptotic cells.

### 3.3. Chronic Inflammation and B Cell Subtypes

Correlation between B lymphocyte subtypes and inflammatory indices, namely CRP and serum albumin levels, are shown on Table 5. 

### 3.4. Effect of B Cell Alterations on Immune Deficiency in ESRD-T0 Patients

Correlation between B lymphocyte subtypes and serum immunoglobulin levels are described on Table 6.

### 3.5. Correlation between Alterations in B Cell Immunity and Secondary Hyperparathyroidism

The serum phosphate levels showed a significant positive correlation with the percentage of CD19+ cells (r = 0.4, *p* = 0.01) and a negative correlation with the percentage of early apoptotic CD3+ lymphocytes (r = −0.4, *p* = 0.02), as well as with the number of early apoptotic CD3+ lymphocytes (r = −0.4, *p* = 0.03) and early apoptotic B lymphocytes (r = −0.4, *p* = 0.03). The patients with secondary hyperparathyroidism had significantly reduced proportions of CD19+BAFF+ and CD19+IgM+ cells (Figure 3).

### 3.6. Changes after the Initiation of Dialysis

Eighteen patients started on CAPD, and twenty-two on HD, with the treatment decision based upon patients’ preference and social situations. No differences in clinical, biochemical and immunological data were evident between them; the demographic and laboratory data at T0 (the starting day on the preferred dialysis method) are shown in Appendix A.

#### 3.6.1. White Blood Cell Count and Subpopulations

Six months later, at T6, the white blood cell count remained stable in those patients started on CAPD, from 6400 (1818) to 6400 (2450) K/μL, *p* = NS, despite the slight reduction of neutrophils from 4000 (1425) to 3850 (1900) K/μL, *p* = NS. The total lymphocyte numbers were significantly increased from 1350 (710) to 1550 (675) K/μL, *p* = 0.01.

On the other hand, the patients who started on HD showed a significant reduction of WBC, from 7900 (3675) to 6600 (2825) K/μL, *p* = 0.002, mainly attributed to the reduction of neutrophils, from 5762 (3375) to 4200 (2275) K/μL, *p* = 0.006. The total lymphocyte numbers were slightly but not significantly increased, from 1250 (1118) to 1500 (600), *p* = NS (Figure 4).

#### 3.6.2. B Lymphocytes

The changes in B cell and subtypes are described in Table 7.

Both dialysis modalities significantly reduced the percentage of CD19+ lymphocytes. The innate B1 cells also declined in patients subjected to both methods, leading to a reduction of the ratio innate B1/conventional B2 cells, while the percentage of naïve B cells increased, and the numbers of CD19+BAFF+ and CD19+IgM+ cells did not show significant changes.

#### 3.6.3. Changes in the Number of Apoptotic Cells

The percentage of early apoptotic lymphocytes was reduced, while those of late apoptotic lymphocytes and B cells increased in all patients, irrespective of the type of treatment, CAPD or HD, during the 6 months of follow-up, as described in Table 8 and depicted in Figure 5.

## 4. Discussion

In the present study, we evaluated the effect of ESRD, at the pre-dialysis stage, on the B lymphocyte compartment, in terms of phenotypic alterations, in comparison with healthy controls and, subsequently, prospectively assessed the influence of dialysis. The patients were initially assessed at the time of dialysis start and then after 6 months on either HD or CAPD. Thus, the specific effects of chronic kidney disease at the pre-dialysis stage on the B cell immune phenotype and the further changes after starting dialysis were evaluated. We also compared the different effects of HD and CAPD on the expression of specific B lymphocyte surface molecules.

The surface B lymphocytes receptors studied were CD5, CD27, IgM and BAFF. CD19+CD5+ cells are defined as B1 or innate B cells, CD19+CD27− cells as naïve, CD19+IgM+ cells as non-switched, CD19+CD27+ cells as memory and CD19+BAFF+ cells as proliferative B cells. The percentages and ratios of the subpopulations were estimated, as well as their exact counts at T0 and T6. At the same time points, annexin expression in B lymphocytes, indicating apoptotic cells, was also evaluated [16,17,18].

The numbers of total lymphocytes, total B cells and most of the subtypes measured were significantly reduced in ESRD patients at the pre-dialysis stage, compared to age- and gender-matched healthy controls. CD19+CD5+ cells were the only cell type which was significantly increased in ESRD-T0 patients, demonstrating the presence of a clear shift towards innate B1 cells. This was also evident by the significant reduction of B2 cells and the increased B1/B2 cell ratio in ESRD patients at T0, compared to controls.

Previous research performed by us and other institutions demonstrated an undoubtedly significant lymphopenia in ESRD patients, even at pre-dialysis stages, including B cell lymphopenia [3,19,20]. The distribution of B cell subpopulations within the whole B cell cohort was also of great importance. Although for the T cell compartment there is a strong evidence of a senescent phenotype predominance [12,13], for B lymphocytes this has not been proved so far, indicating that B lymphocytes in ESRD cannot be simply characterized as senescent, but they acquire a different and yet poorly defined phenotype [12].

In the present study, our choice to evaluate CD5 expression in B lymphocytes was based on its utility to distinguish the B cell compartment into two subtypes, namely, innate B1 cells and conventional B2 cells. The innate B1 cells constitute a specific B cell cluster, which seems to have properties of both the innate and the adaptive immune system. Together with marginal zone B cells and innate lymphoid cells, they both exhibit innate-like effects, including antigen internalization, regulatory B cell and helper T cell functions, and retain their capacity to produce cytokines and natural antibodies and activate the classical complement pathway and, thus, provide an early defense against bacterial and viral infections after birth. Accordingly, these lymphocyte subtypes are regarded to stand on the borderlines between innate and adaptive immunity and have been characterized very precisely to belong to the innate compartment of the adaptive immune system [21,22,23].

To our knowledge, this is the first study in the literature that assessed the expression of innate B1 and conventional B2 cells in patients with ESRD. The role of innate B1 and conventional B2 cells has recently been estimated in predicting mortality in pre-dialysis chronic kidney disease elderly patients. In addition to B cell lymphopenia which has an established impact on cardiovascular and all-cause mortality in ESRD patients, the reduced levels of both B1 and B2 cells were also associated with a declining renal function and with mortality [24,25,26].

B-1 cells act as the front line of defense against microbial or viral infections, by spontaneously secreting natural antibodies and cytokines, such as IL-10 and GM-CSF, and neutralizing pathogens before the launch of the adaptive immune response. The majority of resting serum IgM, almost 80–90%, and IgA, about 50%, are secreted from B1a cells [21,22]. In accordance with this effect, our results showed a significant positive correlation of the B1 cell population with the serum levels of IgM and IgG. B1 cells have been found to increase after infection with influenza virus or S. pneumonia and also in chronic inflammation, such as in the presence of inflammatory bowel disease, experimental autoimmune encephalomyelitis and atherosclerosis. [27] The reason B1 cells were found increased in our ESRD patients is obscure; we can only anticipate that certain metabolic conditions which predominate in the presence of renal failure may accelerate the production of innate cells’ immunity. In addition, a significant reduction of B lymphocytes could also stimulate the generation of B1 cells, which derive from different progenitors found in the bone marrow [27,28,29].

B cells were also subdivided, according to their CD27 expression, in CD19+CD27− and CD19+CD27+ cells, defined as naïve and memory cells, respectively. Both subpopulations were significantly reduced in the ESRD patients at T0, compared to the controls. In order to estimate the preponderance of naïve or memory cells, we calculated the CD19+CD27−/CD19+CD27+ cells ratio. There was no significant difference between patients and controls, confirming previous findings that B lymphocytes are reduced in chronic kidney disease but are not transformed towards a senescent phenotype [12]. Memory B cells derive from naïve cells which, after recognizing an antigen, undergo clonal expansion and differentiation. Several previous studies showed a reduction in memory B cells, responsible for the susceptibility to infections and reduced response to vaccination [12,19,20,27,30].

In our study, there was a significant positive correlation between memory B cells and IgA and IgM serum levels and a negative correlation between memory and CD19+BAFF+ cells and CRP levels. Although there are no previous reports in the literature estimating the correlation between B lymphocyte subpopulations and serum inflammatory markers and immunoglobulin levels, our results are in agreement with the function of specific B lymphocyte subpopulations and indicate that B cells retain their own aptitudes and skills in the presence of end-stage renal disease.

B lymphocytes expressing the BAFF receptor and IgM were significantly reduced in our study population, in accordance with several previous findings [19,31]. The reduction of CD19+BAFF+ cells has been previously associated with an increase in serum BAFF levels. BAFF is a cytokine that enhances B cell survival and proliferation, its production being stimulated by the presence of chronic inflammation and lymphopenia [19,32].

In ESRD, the circulating BAFF levels may be the result of chronic inflammation and act as a compensation factor for lymphopenia. However, this elevation seems to be followed by adverse effects, as it has been associated with an increased risk of acute antibody-mediated rejection in renal transplant recipients [33,34].

Six months following the initiation of dialysis, the WBCs were significantly reduced in the HD patients, mainly due to neutrophil reduction, while the lymphocyte number remained stable. In the CAPD patients, a slight, yet significant increase in lymphocytes was evident. The percentage of CD19+ cells was reduced in both groups, with no significant difference in their absolute numbers. Interestingly, the innate B1 cells were invariant in both groups, while naïve, non-switched and proliferative B lymphocytes were diminished in the HD group only. Overall, there were no significant changes after the initiation of dialysis, and also, there were no important differences depending on the two dialysis methods. The prospective effect of the dialysis methods on B lymphocytes was not investigated before. We extensively studied the effect of HD and CAPD on T lymphocytes, which revealed remarkable differences between the two methods, with our results supporting CAPD use in terms of reinstating the T cell phenotype [13,14,15].

The percentage of apoptotic lymphocytes and B cells was increased in ESRD-T0 patients compared to the controls. Moreover, the ratio of early/late apoptotic lymphocytes and B cells was significantly different between ESRD-T0 patients and controls, with the phenomenon of late apoptosis predominating in ESRD patients. After 6 months on dialysis, the percentage of early apoptotic lymphocytes was significantly reduced in all patients irrespective of the dialysis method; thus, the ratio of early/late apoptotic cells was reduced, meaning that neither method reduced the number of late apoptotic lymphocytes. There are only few studies in the literature that evaluated the apoptosis of lymphocytes in the presence of ESRD. They all agree that the rate of immune cell apoptosis is increased in the presence of chronic kidney disease and affects polymorphonuclear leukocytes, monocytes and lymphocytes both in children and in adults. Some investigators suggest that the profound lymphopenia in ESRD may be the result of accelerated apoptosis, due to uremic toxin accumulation [17,35,36].

## 5. Conclusions

In the present study, we showed a significant reduction of B lymphocytes in ESRD patients even at the pre-dialysis stage, compared to healthy controls. The reduction affected conventional B2 cells, naïve, memory, CD19+BAFF+ and CD19+IgM+ cells. We presented a significant increase in the proportion of innate B1 cells, something that had not been described before and may be the result of B lymphocyte downregulation due to uremia. Finally, the apoptotic phenomenon affecting lymphocytes and B cells was prominent in our patients, involving mainly late apoptotic lymphocytes, and seemed to be further accelerated after starting HD. 

## Figures and Tables

**Figure 1 life-13-00860-f001:**
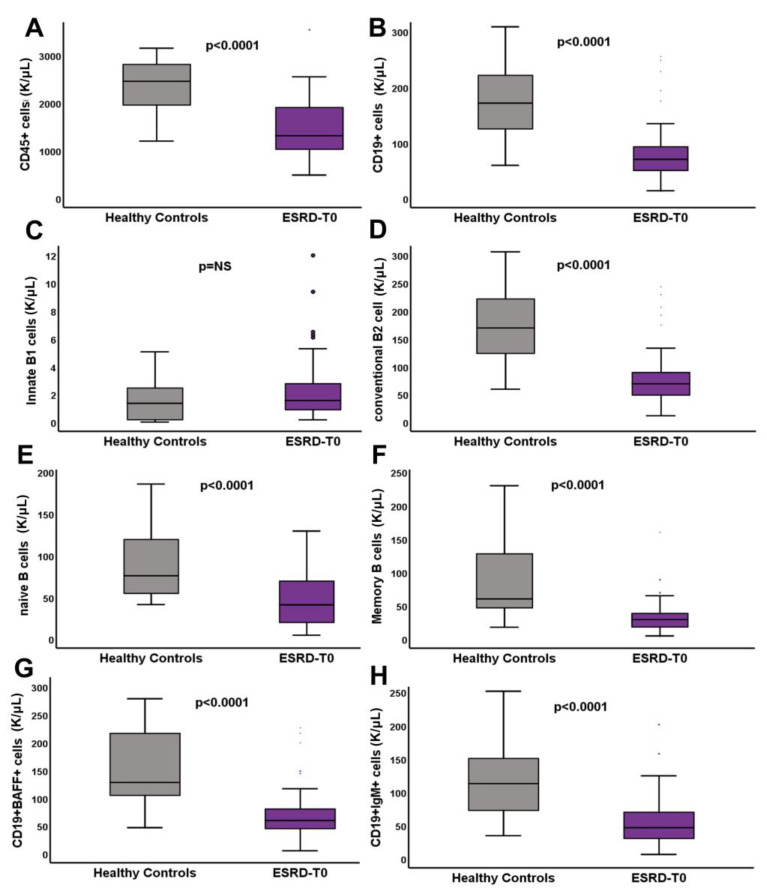
Differences in CD45+ cells (**A**), CD19+ (**B**) innate B1 (**C**), conventional B2 (**D**), naïve B (**E**), memory B (**F**), CD19+BAFF+ (**G**) and CD19+IgM+ (**H**) cells (K/μL) between ESRD-T0 patients and healthy controls.

**Figure 2 life-13-00860-f002:**
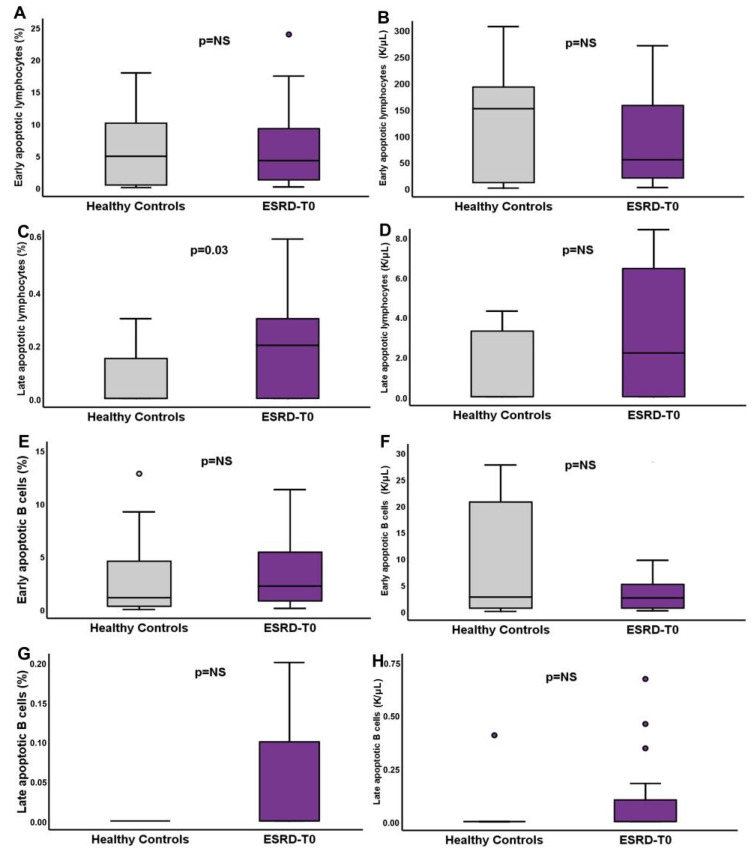
Differences in apoptotic lymphocytes and apoptotic B cells between ESRD-T0 patients and healthy controls. Percentages and counts of early apoptotic lymphocytes, (**A**,**B**), late apoptotic lymphocytes, (**C**,**D**), early apoptotic B cells (**E**,**F**) and late apoptotic B cells (**G**,**H**).

**Figure 3 life-13-00860-f003:**
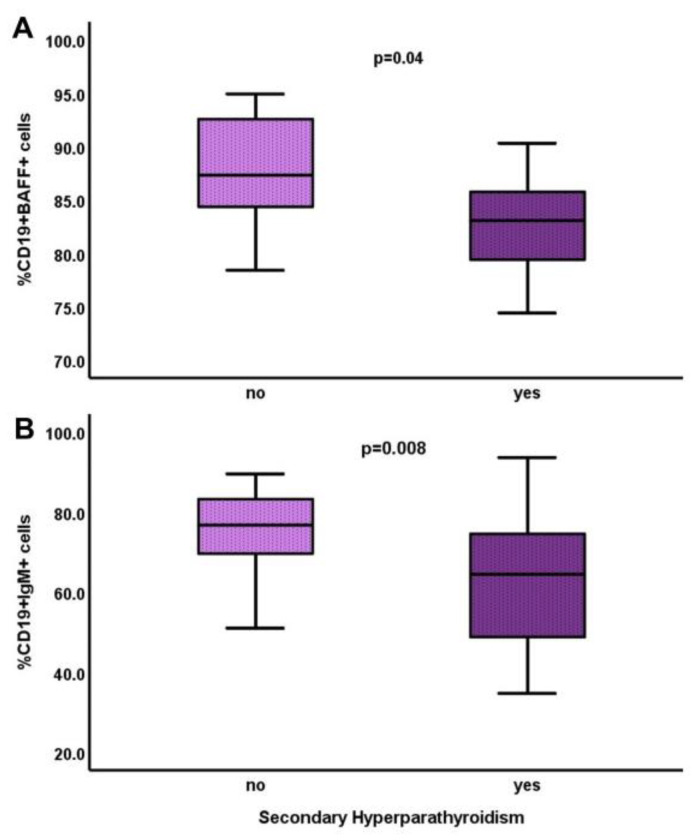
Differences in the percentages of CD19+BAFF+ (**A**) and CD19+IgM+ (**B**) cells between ESRD-T0 patients with and without secondary hyperparathyroidism.

**Figure 4 life-13-00860-f004:**
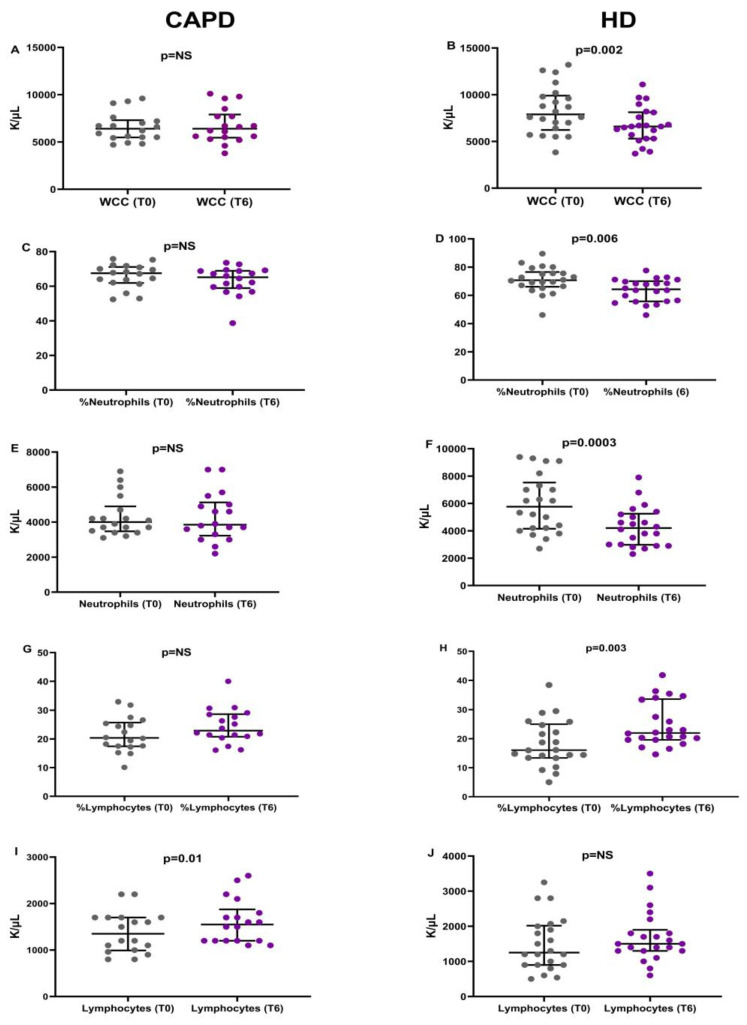
Alterations in the total numbers and percentages of WBC and their subtypes from T0 to T6. Changes in the WBC count (**A**,**B**), percentage (**C**,**D**) and total number of neutrophils (**E**,**F**) and percentage (**G**,**H**) and total number (**I**,**J**) of lymphocytes in patients started on CAPD or HD, respectively.

**Figure 5 life-13-00860-f005:**
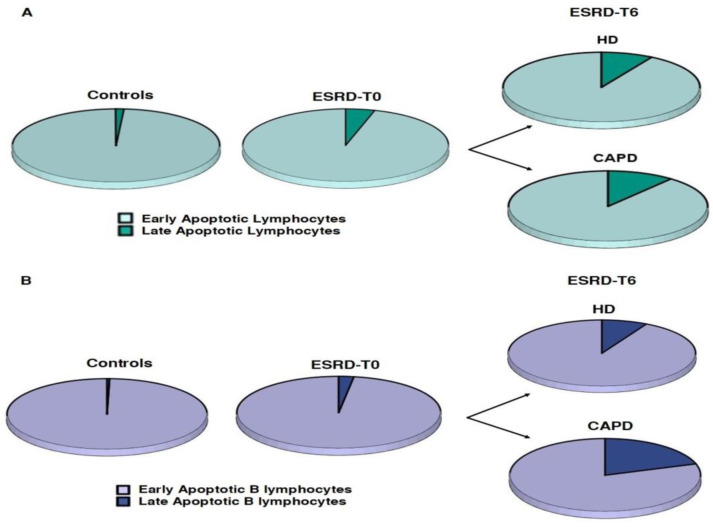
Changes in the ratio of early/late apoptotic CD3+ lymphocytes (**A**) and B lymphocytes (**B**) between controls and ESRD-T0 patients, with further alterations after 6 months on HD or CAPD.

**Table 1 life-13-00860-t001:** Hematology and biochemistry parameters in ESRD-T0 patients and healthy controls.

	Healthy Controls	ESRD-T0	*p*-Values
n	20	40	
Age (yrs)	54.00(15)	60.00(15)	NS
WBC (K/μL)	6900.00(2150)	7015.00(3575)	NS
Neutrophils (%)	50.40(16.1)	69.5(10.7)	<0.0001
Lymphocytes (%)	36.10(12.5)	18.45(10.7)	<0.0001
NLR	1.3(1.01)	3.6(3.5)	<0.0001
Monocyte (%)	8.50(2.8)	7.8(2.1)	NS
Eosinophils (%)	2.90(2.4)	2.4(3.5)	NS
Basophils (%)	0.40(0.2)	0.5(0.3)	NS
Neutrophils (K/μL)	3200.00(1300)	4300.00(2675)	<0.0001
Lymphocytes (K/μL)	2500.00(905)	1250.00(860)	<0.0001
Monocytes (K/μL)	500.00(190)	550.00(275)	NS
Eosinophils (K/μL)	200.00(210)	200.00(200)	NS
Basophils (K/μL)	0.00(25)	0.00(100)	NS
CRP	0.4(0.2)	1.8(8.1)	NS
Serum Total Protein (g/dL)	7.6(4.5)	6.7(1)	0.01
Serum Albumin (g/dL)	4.3(1.2)	3.6(0.8)	<0.0001
Intact PTH	24.9(12	284(561)	<0.0001
Serum Ca (mg/dL)	9.5(1.5)	8.4(1.2)	<0.0001
Serum P (mg/dL)	3.9(0.5)	5.5(1.8)	<0.0001
Cholesterol (mg/dL)	179.00(159)	166.00(61)	0.02
Triglycerides (mg/dL)	86.00(78)	140.00(94)	0.04
LDL (mg/dL)	102.00(48)	90.00(36)	0.003
HDL (mg/dL)	61.00(38)	40.50(10)	0.009
Troponin (mg/dL)	0(1.5)	0.12(8.5)	0.001

Abbreviations: ESRD-T0: End-Stage Renal Disease at T0, WBC: White Blood Cells, CRP: C-Reactive Protein, PTH: Parathyroid hormone, LDL: Low-Density Lipoprotein, HDL: High-Density Lipoprotein, NS: Non-Significant.

**Table 2 life-13-00860-t002:** Differences in immunological indices between ESRD-T0 patients and controls.

	Healthy Controls	ESRD-T0	*p*-Values
	20	40	
C3 (mg/dL)	94.1(40.7)	80(21.2)	0.04
C4 (mg/dL)	22.5(9.2)	21.8(7.3)	NS
IgA (mg/dL)	154(145)	216(106)	NS
IgG (mg/dL)	935(347)	1020(512)	NS
IgM (mg/dL)	81.75(86)	60.3(66)	NS

**Table 3 life-13-00860-t003:** B lymphocytes and their subpopulations, percentage and absolute numbers in ESRD-T0 patients and healthy controls.

	Healthy Controls	ESRD-T0	*p*-Values
n	20	40	
CD45+ (%)	36.7(13.4)	2.9(9.8)	<0.0001
CD45+ (K/μL)	2451.8(1081)	1312(925.8)	<0.0001
B Lymphocytes (CD19+) (%)	7.6(4.5)	5.9(5.6)	0.12
B Lymphocytes (CD19+) (K/μL)	171(249)	70.8(46/5)	<0.0001
Innate B1 cells (CD19+CD5+) (%)	0.6(1.1)	2.7(3.7)	<0.0001
Innate B1 cells (CD19+CD5+) (K/μL)	1.3(2.4)	1.5(1.8)	0.14
Conventional B2 cells (CD19+CD5−) (%)	99.4(1.1)	97.1(4.5)	<0.0001
Conventional B2 cells (CD19+CD5−) (K/μL)	168.9(106)	68.6(43)	<0.0001
Ratio B1/B2 cells	0.006(0.01)	0.02(0.2)	<0.0001
Naive B cells (CD19+CD27−) (%)	42.5(25.6)	39.7(38.1)	0.26
Naive B cells (CD19+CD27−) (K/μL)	59.7(88.4)	31.2(22.1)	<0.0001
Memory cells (CD19+CD27+) (%)	57.5(15.3)	60.3(38.1)	0.36
Memory cells (CD19+CD27+) (K/μL)	84.3(78.1)	42.1(63.6)	0.002
Ratio Naïve/Memory cells	1.3(1.2)	1.8(1.9)	0.27
CD19+BAFF+ cells (%)	89.7(16.1)	84.1(9)	0.89
CD19+BAFF+ cells (K/μL)	127.9(123.7)	59.7(37.8)	<0.0001
CD19+IgM+ cells (%)	58.2(22.1)	69(26.2)	0.46
CD19+IgM+ cells (K/μL)	112.5(81.7)	48.9(42.8)	<0.0001

**Table 4 life-13-00860-t004:** Differences in the percentages and absolute numbers of apoptotic CD3+ lymphocytes and B lymphocytes between ESRD-T0 patients and controls.

	Controls	ESRD-T0	*p*-Values
n	20	40	
Apoptotic Lymphocytes			
Early Apoptotic Lymphocytes (%)	4.9(9.8)	3.6(2)	0.8
Early Apoptotic Lymphocytes (K/μL)	150(36)	45.7(142)	0.5
Late Apoptotic Lymphocytes (%)	0.08(0.02)	0.25(0.1)	0.02
Late Apoptotic Lymphocytes (K/μL)	1.9(0.7)	2.4(6.4)	0.2
Ratio Early/Late Apoptotic Lymphocytes	73.7(54)	32(41)	0.01
Apoptotic B cells			
Early Apoptotic B cells (%)	1.1(4.4)	2.4(1.4)	0.2
Early Apoptotic B cells (K/μL)	2.7(3.8)	2.6(5)	0.5
Late Apoptotic B cells (%)	0.01(0.009)	0.1(0.4)	0.1
Late Apoptotic B cells (K/μL)	0.09(0.1)	0.1(0.03)	0.2
Ratio Early/Late apoptotic Β cells	110(25.4)	16.8(10.9)	0.03

**Table 5 life-13-00860-t005:** Correlation of B cells subtypes with inflammatory indices, CRP and serum albumin levels, in ESRD-T0 patients.

	CRP		Serum Alb	
	r	*p*-Values	r	*p*-Values
CD19+ cells (K/μL)		NS		NS
Innate B1 cells (K/μL)		NS		NS
Conventional B2 cells (K/μL)		NS		NS
Naïve cells (K/μL)		NS		NS
Memory cells (K/μL)	−0.4	0.003		NS
CD19+BAFF+ cells (K/μL)		NS	−0.32	0.04
CD19+IgM+ cells (K/μL)		NS		NS

**Table 6 life-13-00860-t006:** Correlations of B cell subtypes with serum immunoglobulin levels in ESRD-T0 patients.

	IgA		IgG		IgM	
	r	*p*-Values	r	*p*-Values	r	*p*-Values
CD19+ cells (K/μL)		NS		NS		NS
Innate B1 cells (K/μL)		NS	0.4	0.02	0.37	0.02
Conventional B2 cells (K/μL)		NS		NS		NS
Naïve cells (K/μL)		NS		NS		NS
Memory cells (K/μL)	0.4	0.017		NS	0.4	0.02
CD19+BAFF+ cells (K/μL)		NS		NS		NS
CD19+IgM+ cells (K/μL)		NS		NS		NS
Apoptotic Lymphocytes (K/μL)		NS	0.6	0.003		NS
Early Apoptotic Lymphocytes (K/μL)		NS	0.6	0.001		NS
Late Apoptotic Lymphocytes (K/μL)	−0.44	0.04		NS		NS
Apoptotic B cells		NS	0.5	0.02	0.5	0.01
Early Apoptotic B cells (K/μL)		NS	0.5	0.01	0.5	0.01
Late Apoptotic B cells (K/μL)		NS		NS		NS

**Table 7 life-13-00860-t007:** Changes from T0 to T6 in B Lymphocytes and their subtypes, after 6 months on CAPD or HD.

	CAPD			HD		
	T0	T6	*p*-Values	T0	T6	*p*-Values
CD19+ cells (%)	5.1(5.1)	3.3(4.6)	0.05	6.3(5.8)	4(2.4)	0.05
CD19+ cells (K/μL)	60.6(64.9)	59.1(60)	NS	77.2(35.2)	68(48.4)	NS
Innate B1 cells (%)	2.6(4)	1.5(4)	NS	3.2(5.4)	1.7(2)	0.02
Innate B1 (K/μL)	1.4(1.5)	0.77(1.2)	NS	2.3(4.9)	0.9(1)	0.004
Conventional B2 cells (%)	97.3(3.9)	98.4(4.1)	NS	96.7(5.3)	98.3(2)	0.02
Conventional B2 cells (K/μL)	58.9(51.4)	55.4(57.8)	NS	74.6(26.4)	60.1(56.7)	NS
Ratio B1/B2 cells	0.02(0.04)	0.01(0.04)	NS	0.03(0.06)	0.01(0.02)	0.03
Naive cells (%)	63.2(36)	96(4.1)	<0.0001	64.6(32.1)	95.6(4.3)	<0.0001
Naive cells (K/μL)	30.5(54.9)	30(52)	NS	43.3(46.7)	29.9(50.3)	NS
Memory cells (%)	36.7(36)	43(39.7)	NS	35.3(32.2)	46.6(32.9)	NS
Memory cells (K/μL)	23.6(33.3)	25.8(37)	NS	30.1(14.4)	27.1(45.2)	NS
Ratio naive/memory cells	1.7(1.9)	2.2(2.1)	0.01	1.8(2.2)	2(1.8)	NS
% CD19+BAFF+ cells	85.6(7.1)	88.6(16.8)	NS	83.4(11.8)	77.3(23.2)	NS
CD19+BAFF+F cells (K/μL)	51.7(58)	50.3(52)	NS	62.1(28.3)	38.3(48.4)	NS
% CD19+IgM+ cells	66.7(26.1)	69.2(25.2)	NS	70(18.1)	66.2(30.1)	NS
CD19+IgM+ cells (K/μL)	35.6(43)	30.7(60.2)	NS	56(36)	37.7(41.3)	NS

**Table 8 life-13-00860-t008:** Changes in apoptotic lymphocytes and apoptotic B cells during 6 months on CAPD or HD.

	CAPD			HD		
	T0	T6	*p*-Values	T0	T6	*p*-Values
Early Apoptotic Lymphocytes (%)	2.8(4.9)	3.4(6.2)	0.01	7.4(13)	2.7(11)	0.002
Early Apoptotic Lymphocytes (K/μL)	37.4(68.2)	63.6(124)	0.9	110(196)	35(114)	0.5
Late Apoptotic Lymphocytes (%)	0.3(0.4)	0.05(0.8)	0.7	0.2(0.3)	0.3(0.6)	0.01
Late Apoptotic Lymphocytes (K/μL)	3(6.9)	0.6(4.8)	0.5	1.2(5.7)	4.2(7.2)	0.02
Early Apoptotic B cells (%)	2(4.5)	1.5(2.8)	0.1	4(5.4)	2.9(5.3)	0.8
Early Apoptotic B cells (K/μL)	1.7(2.8)	1.2(3)	0.2	3.5(5.5)	4.2(6.5)	0.8
Late Apoptotic B cells (%)	0.13(0.2)	0.1(0)	0.08	0.09(0.2)	0(0)	0.2
Late Apoptotic B cells (K/μL)	0.08(0.1)	0(0)	0.2	0.12(0.3)	0(0)	0.09

## Data Availability

Research data are available on request.

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
