# Peer review of "Prospective Analysis of B Lymphocyte Subtypes, before and after Initiation of Dialysis, in Patients with End-Stage Renal Disease"

_life, 2023, doi:10.3390/life13040860_

Round 1
Reviewer 1 Report
Daikidou D and colleagues report a phenotypic study of different B cell subpopulations in a small cohort of ESRD patients. the study is of interest since adds some details to the B cell dysfunction in ESRD and dialysis patients.
nonetheless to improve the interpretation of the results some important data are needed:
original nephropathy and possible lymphodepletive/steroids therapies
response to vaccinations
other clinical informations as comorbidities
these elements could affect the interpretation of the results
extensive revision of the manuscript is needed to improve the overall readability.
Author Response
Thank you very much for revising our manuscript, your comments helped us to substantialy improve it.
Here are the answers to your comments
Reviewer 1
Daikidou D and colleagues report a phenotypic study of different B cell subpopulations in a small cohort of ESRD patients. the study is of interest since adds some details to the B cell dysfunction in ESRD and dialysis patients.
nonetheless to improve the interpretation of the results some important data are needed:
original nephropathy and possible lymphodepletive/steroids therapies
response to vaccinations
Thank you for this comment, we have added a paragraph with information regarding primary disease. According to the protocol, no patient had steroid and immunosuppressive treatment, and also, infection or systemic disease during the last 6 months. The study contacted before vaccination against COVID-19, therefore, the response was not applicable.
other clinical information as comorbidities these elements could affect the interpretation of the results
Thank you very much, this is a very interesting comment. Comorbidities and clinical situations that could affect immune status are systemic diseases, infections, hematological disorders, or recent immunosuppression. The presence of any of these situations was among the seclusion criteria. Also, and most importantly, we excluded patients who had received monoclonal antibodies against B lymphocytes (Rituximab, Belimumab) in the past.
extensive revision of the manuscript is needed to improve the overall readability.
Thank you so much for your comment, we have extensively revised our manuscript and we believe it is much easier for the reader to follow our results and discussion
Reviewer 2 Report
Manuscript: Prospective analysis of B lymphocyte subtypes, before and after initiation of dialysis in patients with end stage renal disease.
is an interesting paper showing differences in B lymphocyte levels in patients with ESRD compared to control patients.
The work is prepared according to general standards for scientific publication. The methodological description of the work is also the most correct from the criteria for recruiting patients by performing tests and then analyzing the results.
In my opinion, the work is a valuable contribution to the world of science and can be published after correcting minor issues, which I present below:
1. Please check your manuscript again for spaces, periods at the end of a sentence (lines: 61, 207, 279, 346, 350, 375, etc.)
2. tables 5 and 6 are colored, the other tables are not blue, which style do you keep?
3. In the materials and methods related to flow cytometry, please complete the information about the company and the clones of the antibodies used.
4. For additional materials 1, please explain the symbols P, N, O, K under the picture - which subpopulations these symbols refer to.
Author Response
Thank you very much for your time and efforts to improve our paper, we have revised the whole manuscript according to your comments and suggestions,
Here are the answers to your comments
Manuscript: Prospective analysis of B lymphocyte subtypes, before and after initiation of dialysis in patients with end stage renal disease.
is an interesting paper showing differences in B lymphocyte levels in patients with ESRD compared to control patients.
The work is prepared according to general standards for scientific publication. The methodological description of the work is also the most correct from the criteria for recruiting patients by performing tests and then analyzing the results.
In my opinion, the work is a valuable contribution to the world of science and can be published after correcting minor issues, which I present below:
- Please check your manuscript again for spaces, periods at the end of a sentence (lines: 61, 207, 279, 346, 350, 375, etc.)
Thank you very much, we have checked the whole manuscript and corrected these.
- tables 5 and 6 are colored, the other tables are not blue, which style do you keep?
Thank you for this comment, we have changed it
- In the materials and methods related to flow cytometry, please complete the information about the company and the clones of the antibodies used.
Thank you so much for this point, this was really missed, we have completed all information regarding the antibodies we used
- For additional materials 1, please explain the symbols P, N, O, K under the picture - which subpopulations these symbols refer to.
Thank you for your comment, in fact these symbols did not mean anything, we just used them to define the quartiles in space, so we just erased them

Reviewer 3 Report
Work by Daikidou et al. is an extremely interesting literature item, however, it requires a few editorial changes to improve the clarity of the manuscript:
• Line 49-50, please remove the strikethrough;
• Please clearly specify the purpose of the research in the introduction section, it will make it easier for the reader to receive the work;
• Line 58 a space before [10], same as line 61 before [11-13];
• Please explain the abbreviations HD and CAPD at the time of their first use in the text, ie line 62, and remove their expansion in line 77 (leave only the abbreviations);
• Please change Covid-19 to COVID-19 line 86;
• Please explain the abbreviation CRP line 103 and (serum Ca, serum P, intact PTH) line 104;
• Line 114 please explain the abbreviation CKD (as an abbreviation it appears here for the first time);
• Please explain the abbreviations used in tables 1 and 2 below the table;
• In each table, please add p-values instead of just p;
• Please use the target formatting of tables 5 and 6 to be consistent with the journal guidelines and previous tables;
• Please check the discussion section, in many places spaces are missing before cited literature and there are dots before cited literature;
Author Response
Thank you very much for this valuable revision and for your time to read and revise our paper.
We have made all changes, and here are the answers to your comments
Work by Daikidou et al. is an extremely interesting literature item, however, it requires a few editorial changes to improve the clarity of the manuscript:
- Line 49-50, please remove the strikethrough;
Thank you, we have removed it
- Please clearly specify the purpose of the research in the introduction section, it will make it easier for the reader to receive the work;
Thank you very much for this comment, we tried to clearly describe the purpose of this study in the last paragraph of introduction, and hope this makes our manuscript much easier to follow.
- Line 58 a space before [10], same as line 61 before [11-13];
Thank you, we have done this
- Please explain the abbreviations HD and CAPD at the time of their first use in the text, ie line 62, and remove their expansion in line 77 (leave only the abbreviations);
Thank you, we have done this
- Please change Covid-19 to COVID-19 line 86;
Thank you, we have done this
- Please explain the abbreviation CRP line 103 and (serum Ca, serum P, intact PTH) line 104;
Thank you, we have done this
- Line 114 please explain the abbreviation CKD (as an abbreviation it appears here for the first time);
Thank you, we have done this
- Please explain the abbreviations used in tables 1 and 2 below the table;
Thank you, we have done this
- In each table, please add p-values instead of just p;
Thank you, we have done this
- Please use the target formatting of tables 5 and 6 to be consistent with the journal guidelines and previous tables;
Thank you, we have done this
- Please check the discussion section, in many places spaces are missing before cited literature and there are dots before cited literature;
Thank you, we have done this

Round 2
Reviewer 1 Report
Authors have add some notes to the comments required. Although English has been improved, still some errors are present, i.e. the first words of the conclusion sections lacking an "in".